# The Heart as a Target of Vasopressin and Other Cardiovascular Peptides in Health and Cardiovascular Diseases

**DOI:** 10.3390/ijms232214414

**Published:** 2022-11-20

**Authors:** Ewa Szczepanska-Sadowska

**Affiliations:** Department of Experimental and Clinical Physiology, Laboratory of Center for Preclinical Research, The Medical University of Warsaw, Banacha 1B, 02-097 Warsaw, Poland; eszczepanska@wum.edu.pl

**Keywords:** vasopressin, angiotensin, oxytocin, cytokines, heart failure, hypertension, hypoxia, inflammation, resuscitation, stress

## Abstract

The automatism of cardiac pacemaker cells, which is tuned, is regulated by the autonomic nervous system (ANS) and multiple endocrine and paracrine factors, including cardiovascular peptides. The cardiovascular peptides (CPs) form a group of essential paracrine factors affecting the function of the heart and vessels. They may also be produced in other organs and penetrate to the heart via systemic circulation. The present review draws attention to the role of vasopressin (AVP) and some other cardiovascular peptides (angiotensins, oxytocin, cytokines) in the regulation of the cardiovascular system in health and cardiovascular diseases, especially in post-infarct heart failure, hypertension and cerebrovascular strokes. Vasopressin is synthesized mostly by the neuroendocrine cells of the hypothalamus. There is also evidence that it may be produced in the heart and lungs. The secretion of AVP and other CPs is markedly influenced by changes in blood volume and pressure, as well as by other disturbances, frequently occurring in cardiovascular diseases (hypoxia, pain, stress, inflammation). Myocardial infarction, hypertension and cardiovascular shock are associated with an increased secretion of AVP and altered responsiveness of the cardiovascular system to its action. The majority of experimental studies show that the administration of vasopressin during ventricular fibrillation and cardiac arrest improves resuscitation, however, the clinical studies do not present consisting results. Vasopressin cooperates with the autonomic nervous system (ANS), angiotensins, oxytocin and cytokines in the regulation of the cardiovascular system and its interaction with these regulators is altered during heart failure and hypertension. It is likely that the differences in interactions of AVP with ANS and other CPs have a significant impact on the responsiveness of the cardiovascular system to vasopressin in specific cardiovascular disorders.

## 1. Introduction

Effective function of the heart is necessary for circulation of the blood in the cardiovascular system in a manner adjusted to actual needs. Action of the heart depends on the automatism of cardiac pacemaker cells, which is precisely tuned by the autonomic nervous system (ANS), and multiple endocrine and paracrine factors. The cardiac muscle forms a functional syncytium, allowing the rapid transmission of signals that are generated in the cardiac conduction system. Under physiological conditions, cells of the cardiac muscle are excited by action potentials generated in the sinoatrial node and transmitted to the atrioventricular node and other parts of the conductive system. In cardiovascular diseases, they may also be initiated in other cardiac cells, serving as ectopic pacemakers.

It is well known that the ANS plays a crucial role in the adaptation of the cardiovascular system to challenges associated with stress, anxiety, ischemia, pain, inflammation and other pathologies. Under resting conditions, the heart and the coronary vessels receive excitatory inputs from the presympathetic neurons of the ANS, which are located mainly in the rostral ventrolateral medulla (RVLM). The signals are transmitted to the sympathetic ganglia of the spinal cord, and subsequently to the heart and vessels. In addition, the heart is innervated by the parasympathetic projections from the nucleus ambiguous and the dorsal motor nucleus of the vagus (DMVNc, AmbNc). Efficient coordination of abundant excitatory and inhibitory inputs from the medullary and supramedullary regions occurs in the RVLM and the nucleus of the solitary tract (NTS) of the brainstem [1,2]. The dysregulation of the interplay between the sympathetic and parasympathetic components of the ANS plays an important role in the generation of the cardiac arrhythmias. The destruction of connections between the supramedullary regions and the NTS significantly reduces baseline activity of the NTS neurons and diminishes their activation by the afferent projections [3,4].

The endogenous cardiovascular peptides (CPs), particularly vasopressin, angiotensins, oxytocin and cytokines, belong to essential paracrine factors affecting the function of the heart. They may be synthesized either in the heart itself or in other organs, including the brain, the kidneys and the digestive system, and have access to the cardiac cells from systemic circulation [5,6,7,8,9]. In the heart, the cardiovascular peptides operate through direct actions exerted on cells of the heart and coronary vessels, and indirectly, through modulation of the activity of the ANS and through interaction with other neuroendocrine and humoral factors. At the cellular level, the cardiovascular peptides act by means of specific receptors and a wide range of intracellular mechanisms, engaging the transport of ions, cellular metabolic processes and the release of neurotransmitters [1,2,9,10]. Plurality of these effects enables precise adjustment of cardiac functions to different challenges.

The purpose of the present review is to draw attention to the specific role of vasopressin (AVP) in the regulation of cardiac functions in health and disease and the interaction of vasopressin with other CP. The present survey is focused on the cooperation of vasopressin with angiotensins (Ang), oxytocin (OT) and cytokines, because thus far the interaction of these peptides in the heart has been most intensely explored.

## 2. General Overview of the Vasopressin System

Vasopressin synthetizing, cells, vasopressin peptides and vasopressin receptors belong to the vasopressin system (VPS), which regulates blood pressure, water and electrolyte balance, as well as sensitivity to stress and other emotional challenges [8,9]. Most mammal species release arginine vasopressin, and only some of them use lysine vasopressin (LVP). The AVP gene is composed of exons encoding the sequence of a 145-amino acid polypeptide precursor composed of an N-terminal signal peptide, a sequence of vasopressin (the chief component), a sequence of neurophysin II (NPII) and a sequence of a C-terminal peptide, copeptin. Copeptin, which is a 39-amino acid glycopeptide, is released in equimolar quantities with AVP and is frequently used as a biomarker of vasopressin because it is more stable than the chief peptide [11,12]. Measurements of copeptin have been included in ESC guidelines for the management of non-ST-elevation myocardial infarctions [13].

AVP is synthesized mainly in neuroendocrine cells of the supraoptic nucleus (SON), the paraventricular nucleus (PVN) and the suprachiasmatic nucleus (SCN) of the hypothalamus. In addition, immunocytochemical and functional studies provide evidence for its synthesis in cells of the heart and lungs [14,15,16,17,18,19]. Some studies suggest the presence of a local cardiac vasopressin system [20].

Cellular actions of AVP are mediated by V1a receptors (V1aR), V1b receptors (V1bR) and V2 receptors (V2R), which are located in several organs and tissues with specific representation [5,14,17,21,22,23,24,25,26,27]. In addition, AVP can bind to oxytocin receptors (OTR), whereas oxytocin (OT) interacts with V1aR [28].V1aR have been identified at all levels of the brain and the spinal cord, as well as in the heart, vessels, kidneys, lungs and digestive system (including the liver and the pancreas) [26,29,30,31,32,33]. V1bR have been found in the pituitary, brain, pancreatic gland and lungs [34,35,36]. V2R are located mainly in the kidneys [27,37], however, in the neonate rat V2R were also detected in the heart [17]. In addition, the functional studies provide evidence for the presence of V2R in vessels of the brain, skin and skeletal muscles, as well as in cells releasing Factor VIII [38,39,40]. The cardiovascular and neuroregulatory processes are mediated mainly by V1aR [10,41,42,43,44,45,46,47]. Overexpression of V1aR in mouse heart causes cardiac dysfunction, cardiac hypertrophy, ventricular dilation and overactivation of the Gα_q/11_–mediated pathway [48]. The vasopressin system cooperates closely with the autonomic nervous system in the regulation of blood pressure [46]. The general overview of the vasopressin system is shown in Figure 1.

Multiple studies demonstrate that AVP plays a crucial role in the regulation of cardiovascular homeostasis. AVP secretion is inhibited by increases in blood volume and pressure, and stimulated by hypotension, hypoxia, pain, stress and inflammation—the disturbances that frequently occur during acute and/or chronic cardiovascular diseases. It has been well established that altered regulation of AVP release may have a significant impact on cardiovascular regulation [8,10,22,41,43,44,48,49,50,51,52]. Early studies performed on isolated hearts of dogs showed that the infusion of vasopressin into the left atrium decreased the coronary blood flow and caused myocardial dysfunction [53]. Subsequent investigations provided evidence that intravenous administration of vasopressin in pigs with normal sinus rhythm exerted some positive effects in their cardiovascular systems that were manifested by an increase of the mean arterial blood pressure and elevation of the left anterior descending (LAD) coronary artery cross sectional area. At the same time, the authors noted that application of vasopressin reduced the cardiac index [54,55]. Experiments performed on cardiomyocytes of rats revealed that the stimulation of V1aR causes a dose-dependent increase in myocyte contractile function, [Ca^2+^]_i_ and IP3 [56].

## 3. Role of Vasopressin in Cardiovascular Disturbances

As discussed below, hypoxia, ischemia, pain and stress, which are frequent attributes of cardiovascular diseases, are also effective stimuli for vasopressin release.

### 3.1. Hypoxia and Vasopressin

Many studies provide evidence that hypoxia and/or ischemia provoke the significant release of vasopressin and its surrogate, copeptin [57,58,59,60,61,62]. Moreover, it has been shown that intermittent hypoxia induces direct activation of vasopressinergic neurons in the PVN [63,64] and that its effect is strongly potentiated by the central administration of Ang II [65].

Release of AVP into the systemic circulation during hypoxia is associated with vasoconstriction, which is mediated by V1aR and accounts for generation of significant pressor response [60,66]. However, in some vascular beds (cerebral and pulmonary circulation) AVP can cause vasodilation, which is presumably mediated by the release of nitric oxide [38,67,68]. The pressor effect of AVP during hypoxia is also exerted by the stimulation of the presympathetic neurons located in the PVN and RVLM. The blockade of V1aR within the RVLM modulates the cardiovascular responses evoked by chronic intermittent hypoxia (CIH) and reduces baseline blood pressure in CIH-conditioned rats [69]. It should be noted that acting on V1aR vasopressin modulates function of the carotid chemoreflex and causes a decrease in the respiratory rate [33]. It is likely that the regulation of the carotid chemoreflex by AVP is related to changes of glucose metabolism. Central and systemic application of AVP causes hyperglycemia, which is similar to that observed during hypoxia. Moreover, both the hypoxic and the hyperglycemic responses can be abolished by administration of the V1aR antagonist [70].

There is evidence that AVP plays a positive role in the regulation of the pulmonary blood flow during hypoxia. Chronic administration of AVP in moderate concentration induced a significant reduction of the mean pulmonary arterial pressure and prevented the development of pulmonary hypertension [71].

### 3.2. Role of Vasopressin in Pain and Stress 

Pain frequently informs on pathological processes developing in the cardiovascular system. Strong pain is a symptom of the coronary ischemia and may provoke stress and anxiety that intensify the discomfort of the disease and cause activation of the sympathetic nervous system and the release of cardiovascular peptides [72]. Cardiac pain is transmitted to the spinal cord by sympathetic nociceptors and afferents possessing cell bodies located in the thoracic spinal ganglia. It is also generated in parasympathetic nociceptors conveying impulses by means of vagal afferents to the inferior nucleus of the vagus nerve [73,74].

Pain belongs to non-osmotic factors potently stimulating the release of AVP [74]. Exposure to pain elevates AVP content in perfusates of the PVN and of some other brain structures involved in the regulation of pain (e.g., the periaqueductal gray—PAG; the raphe nuclei; the caudate nucleus—CdN) [72,73,74,75,76,77,78,79]. On the other hand, systemic, intraventricular, intrathecal or topical application of this peptide into specific regions of the brain alleviates pain [72,80,81,82,83,84]. It appears that the analgesic effect of AVP is associated with the stimulation of neurons in the CdN, PAG, raphe magnus nucleus (RMN) and spinal cord, which is caused by the stimulation of V1aR [72,80,81,82,85,86,87]. Systemic administrations of AVP or oxytocin (OT) also exert analgesic effects [72,81]. It is possible that the pain alleviating action of OT is mediated by V1aR because it is mimicked by AVP and can be completely blocked by the V1aR antagonist (SR49059), but not by the OT receptor antagonist (L-368899). The analgesic effect of AVP requires the activation of acid-sensing ion channels in the dorsal root ganglia and it cannot be induced in V1aR knockout mice [88].

Neurogenic stress is frequently present in cardiovascular diseases, especially in cardiac ischemia, and has a significant negative effect on the course of the disease. Experimental studies show that the neurogenic stress provokes the release of AVP, and that the stimulation of V1aR by AVP plays a significant role in the potentiation of the magnitude of the cardiovascular and behavioral responses to stress in hypertension and heart failure [43,44,45,89,90].

## 4. Role of Vasopressin in Cardiovascular Diseases

### 4.1. Vasopressin in Myocardial Infarction

A myocardial infarction (MI) stimulates VPS, causing the significant activation of vasopressinergic magnocellular neurons in the SON and the elevation of plasma AVP levels [91,92,93]. Studies on human beings revealed that acute MI results in a significant increase in the concentration of vasopressin and/or copeptine in blood samples collected from patients with acute myocardial infarction [94,95,96,97,98]. In addition, experiments performed on rats showed that heart failure causes activation of the cardiac vasopressin system [20].

Experimental studies suggest that the elevated release of AVP during cardiovascular disturbances may play a positive role during the recovery from ischemia. For instance, in the rat model of cardiac ischemia-reperfusion injury, intravenous administration of AVP prevented the post-ischemic bradycardia and diminished the incidence of ventricular arrhythmia. Furthermore, administration of AVP reduced the size of the infarct and decreased the expression of some biochemical parameters, which are used as measures of cardiac ischemia (lactate dehydrogenase—LDH; creatine kinase-MB—CK-MB). These effects were significantly reduced by the administration of SR49059, which is a V1R antagonist [99]. In the porcine model of ischemic ventricular fibrillation induced by occlusion of the left coronary artery, the administration of AVP more effectively augmented the coronary perfusion pressure than the administration of epinephrine [100]. On the other hand, in experiments on dogs, it was found that the ischemia of the left ventricular myocardium is associated with enhanced contractile response of the coronary microvessels to AVP [101]. Thus, the mechanism for action of VPS in the post-infarct state is not yet sufficiently elucidated, and it is likely that AVP may engage different processes, depending on its concentration and interaction with other factors.

It appears that the influence of vasopressin on the heart is significantly altered in diabetes mellitus, as it has been shown that AVP exerts a greater vasoconstrictive effect in coronary vessels obtained from patients with diabetes mellitus who underwent cardioplegic arrest and cardiopulmonary bypass than in vessels obtained from patients undergoing the same procedures but not suffering from diabetes [102]. The enhanced contractility of the coronary vessels in diabetic patients was mediated by increased stimulation of V1aR because it was significantly diminished in the presence of the specific V1aR antagonist (SR 4059). The diabetic patients also had a higher expression of V1aR in their atrial tissue samples [102].

### 4.2. Vasopressin in Cardiovascular Shock and Cardiopulmonary Resuscitation

Several studies suggest that the administration of vasopressin may exert beneficial effects during the recovery period in some forms of heart failure. Studies performed on the porcine model of cardiac arrest induced by ventricular fibrillation revealed that vasopressin applied alone, or in association with epinephrine, exerted positive cardiovascular effects, such as elevations of coronary perfusion pressure and cerebral blood flow [103,104]. In studies on pigs exposed to hemorrhagic shock and cardiac arrest administration of vasopressin resulted in prolonged survival, reduced acidosis and improved renal blood flow [105]. In a swine resuscitation model, intravenous administration of AVP a few minutes prior to ventricular fibrillation elicited a significant increase in the mid left anterior descending coronary artery cross sectional area and normalized the sinus rhythm [55].

Studies performed on a porcine cardiopulmonary resuscitation model, in which the animals were treated with different combinations of placebo, AVP and epinephrine, the successful restoration of spontaneous circulation was possible in the group treated with a combination of AVP and epinephrine, but not in the group treated with epinephrine alone [104]. In the same cardiopulmonary resuscitation model, the joined application of epinephrine, AVP and nitroglycerine significantly increased the left ventricular blood flow and global cerebral blood flow. Moreover, combined administration of AVP and epinephrine produced a greater elevation of the cerebral blood flow than the infusion of epinephrine alone [106]. Analysis of data from 7 studies comparing the efficiency of various vasopressors used for the return of spontaneous circulation (ROSC) revealed that AVP significantly improved ROSC during ventricular fibrillation evoked by severe hypothermia [107].

Cardiopulmonary resuscitation in human patients with cardiac arrest is associated with a significant elevation in blood levels for AVP, ACTH, cortisol and renin concentration [108,109]. Positive effects of intravenous administration of 40 U of AVP on the survival of patients with cardiac arrest induced by ventricular fibrillation or vasodilatory shock were reported [110,111,112]. In addition, investigations performed on patients with septic shock revealed that administration of AVP in relatively low doses increases blood pressure and urine output and enhances the responsiveness to catecholamines. However, it should be noted that in some patients with vasodilatory shock, application of vasopressin elicited adverse effects (e.g., Russell [113]). A survey of studies analyzing effects of different vasopressors and their combinations indicated that application of AVP or V1aR agonist (selpressin) permits a reduction in the dose of norepinephrine, which is necessary for cardiovascular stabilization in patients with vasodilatory shocks resulting from sepsis, acute myocardial infarction or cardiovascular surgery [7,113]. AVP and V1R agonist (terlipressin) have been successfully used for the treatment of gastrointestinal bleeding, although some patients responded with arrhythmic complications, manifested by a prolonged QT interval and torsade de pointes [114,115]. Retrospective analysis of adult patients admitted to the medical intensive care unit because of arrhythmia revealed that the early (within 6 h) administration of AVP significantly reduced the new onset arrhythmias and decreased the requirements for catecholamine therapy [116].

A triple-blind randomized trial, analyzing the survival of patients with cardiac arrest admitted to Canadian emergency departments, critical care units and hospitals, did not show an advantage in favor of vasopressin over epinephrine [117]. Similar conclusions can be drawn from a meta-analysis of 1519 patients with cardiac arrest in USA hospitals [118].

## 5. Cooperation of Vasopressin with Other Cardiovascular Peptides

### 5.1. Vasopressin and Angiotensins

The renin-angiotensin system (RAS) acts on the cardiovascular system directly through effects exerted locally in cardiac and vascular cells and indirectly by means of the presympathetic neurons of the brain and sympathetic neurons of the heart [119,120,121,122,123]. In addition, acting on AT1 receptors, Ang II may modulate function of the heart by means of the parasympathetic neurons, as it has been shown that it inhibits the release of Ach induced by vagal stimulation of the left cardiac ventricle [124].

Several studies have shown that cardiovascular disorders significantly influence activity of the RAS and that the cardiovascular effects of the RAS are markedly altered in cardiovascular diseases [10,125,126,127,128,129,130]. The most effective components of the RAS that participate in the cardiovascular regulation include angiotensin II (Ang II), angiotensin III (Ang III), Ang-(1–7), AT1 receptors (AT1R), AT2 receptors (AT2R) and Mas receptors (MasR). The cardiovascular effects of angiotensins may be mediated either by central effects initiated in the brain or through systemic actions exerted in the heart and vessels [10,125,130]. Analysis of cardiac ventricle samples taken during biopsies from patients with stable and unstable angina revealed that the patients with unstable angina synthesize more Ang II and have higher expressions of angiotensinogen, angiotensin converting enzyme (ACE) and AT1R genes. Besides, in cardiac myocytes and fibroblasts, they manifest upregulation of the iNOS gene and inflammatory cytokine genes (TNF-α, IL-6, IFN-γ, TGF-β) [131,132]. It has been shown that knockout of AT1R in mice with an experimental model of aortic constriction that fibrosis and hypertrophy of the cardiac muscle were significantly reduced and incidence of arrhythmia declined [133].

Acute cardiac failure and hypertension cause parallel activation of the angiotensinergic and vasopressinergic systems [10,123]. For instance, occlusion of the left coronary artery in Sprague Dawley rats resulted in a significant increase in plasma concentration of Ang II, AVP and TNF-α [134]. Most likely, the increase of plasma AVP levels in these experiments was mediated by activation of the hypothalamic angiotensin receptors, i.e., knockdown of AT1R in the subfornical organ (SFO) significantly reduced the increase of plasma AVP provoked by coronary constriction [134].

### 5.2. Vasopressin and Oxytocin

Similar to vasopressin, oxytocin is synthesized mainly by neuroendocrine cells of the SON, PVN and SCN, and is released to the systemic circulation in vessels of the posterior pituitary [9,14]. It can also be released by neural projections in the brain and is synthesized in the heart [14,135,136].

As early as in 1964, it was shown that systemic administration of synthetic oxytocin (OT) decreases blood pressure and reduces cardiac arrhythmias in several species, including humans [137]. Similar to AVP and angiotensins, oxytocin regulates the cardiovascular system through direct actions exerted on OT receptors (OTR) located in the heart and vessels, and via indirect effects exerted on neurons of the cardiovascular regions of the brain [14,138]. There is evidence for synthesis of OT and OTR in the heart, vessels and brain and for involvement of OT and OTR in the modulation of blood pressure, heart rate and behavioral responses [51,136,137,138,139,140,141]. In many instances, AVP and OT are secreted together, which enhances the chance of their nonspecific interaction [28,142,143]. Along these lines, it has been shown that local applications of high doses of vasopressin or oxytocin into the central nucleus of the amygdala accelerates HR and magnifies the release of corticosterone [144]. As the responses to AVP and OT could be abolished by pretreatment with a selective oxytocin antagonist, it has been suggested that high concentrations of AVP may regulate cardiovascular responses by means of OTR [144].

Experiments on rats have shown that, acting in the brain, OT reduces cardiovascular responses to stress, and that its buffering action is significantly attenuated after a myocardial infarction [50]. In addition, studies with blockades of OTR and V1aR in the brain revealed significant differences in the interaction between OT and AVP, in regards to these receptors in the regulation of cardiovascular responses to acute neurogenic stress between WKY and SHR rats [51,52]. The SHR responded with significantly greater pressor responses to neurogenic stress than the WKY rats and found that the augmentation could be abolished by intraventricular administration of oxytocin. The latter finding suggested that the enhanced cardiovascular responses of the SHR to stress may partly result from deficient action of oxytocin in the brain [51].

Systemic administration of oxytocin prior to ischemia/reperfusion of the myocardium prevents hypotension during the early phase of ischemia and reduces the infarct size and ventricular arrhythmias in SD rats [145]. It has been also shown that SD rats with a myocardial infarction manifest elevated concentrations of the OTR protein in the cardiac muscle [146].

### 5.3. Vasopressin and Cytokines

Cytokines form a big family of specific, biologically active compounds with proinflammatory or anti-inflammatory properties [147,148,149,150]. Several studies provide evidence that cardiovascular diseases, especially myocardial infarction and cerebrovascular stroke initiate inflammatory and anti-inflammatory processes associated with the release of cytokines [147,148,149,150,151,152,153]. There is also evidence that in a myocardial infarction, cytokines play a particularly important role during the early recovery period [154,155,156,157].

Coronary heart disease and heart failure significantly increase the production of potent inflammatory cytokines (interleukin-1—IL-1; interleukin-6—IL-6; tumor necrosis factor-α—TNF-α) in the heart and other organs [147,148,150,156,157,158,159]. It has also been shown that these cytokines play an essential role in the regulation of blood pressure and cardiac remodeling, through cooperation with angiotensins, vasopressin and nitric oxide [158,159,160,161,162,163,164]. The proinflammatory interleukins cooperate with Ang II and AVP by means of central and systemic effects. Acting in the brain, IL-1β and TNF-α exert pressor effects, which are mediated by Ang II and AT1R [157,159,160]. It has also been shown that the central pressor action of TNF-α is intensified in rats with a myocardial infarction [161]. In the heart, the proinflammatory cytokine TGF-β cooperates with Ang II and this cooperation plays an essential role in the development of cardiac fibrosis [131,162].

There are studies showing that the inflammatory cytokines interact with the vasopressinergic system. Administrations of IL-1β- and IL-6 stimulate release of AVP [163,164,165,166,167,168]. On the other hand, reduced V1aR expression was found in aortic smooth muscle cells, and this was associated with attenuated cardiovascular responsiveness to AVP [168]. In SD rats and murine hearts, AVP was found to induce an expression of IL-6 mRNA and IL-6 protein in fibroblasts [168]. Moreover, experiments on cardiac fibroblasts of neonatal SD rats provide evidence that vasopressin promotes the synthesis of TGF-β1 and collagen through actions exerted on V1aR. It is likely that these effects may play an essential role in the development of myocardial fibrosis [169]. The cardiovascular effects of cytokines are at least partly mediated by nitric oxide [170,171].

Altogether, the present evidence indicates that cytokines, acting directly or in cooperation with other cardiovascular factors, may play an essential role in the adaptation of the cardiovascular system to inflammatory injury and in the reparation of injured tissue.

## 6. Summary and Conclusions

The function of the heart is regulated by the autonomic nervous system and biologically active compounds. The cardiovascular peptides, especially vasopressin, angiotensins, oxytocin and cytokines belong to a family of peptides with a wide range of actions that can regulate hemodynamic parameters through direct actions exerted in cells of the cardiovascular system or indirectly by means of the autonomic nervous system. The cardiac and vascular effects of cardiovascular peptides are significantly altered in cardiovascular disorders induced by a myocardial infarction, hypertension and cardiovascular shock, and by hypoxia, stress, pain and inflammation, which frequently occur in cardiovascular diseases. Putative interaction of vasopressin, angiotensins, oxytocin and cytokines with the autonomic nervous system in the regulation of cardiovascular parameters during stress, anxiety, hypoxia and inflammation is illustrated in Figure 2.

This review is focused on the interaction of vasopressin with the most effective cardiovascular peptides, but it is likely that vasopressin may also interact with other cardiovascular factors. Specifically, the cooperation of vasopressin with aldosterone and pancreatic hormones is worthy of attention [172,173,174,175,176,177].

In conclusion, available evidence indicates that vasopressin effectively interacts with the autonomic nervous system and angiotensin II, Ang(1–7), oxytocin and cytokines in the regulation of the cardiovascular system. Action of these peptides is altered in cardiovascular diseases, as well as during stress, pain and inflammation. Presumably, joint action of these peptides largely determines the responsiveness of the cardiovascular system to pharmacological treatments under pathological conditions.

## Figures and Tables

**Figure 1 ijms-23-14414-f001:**
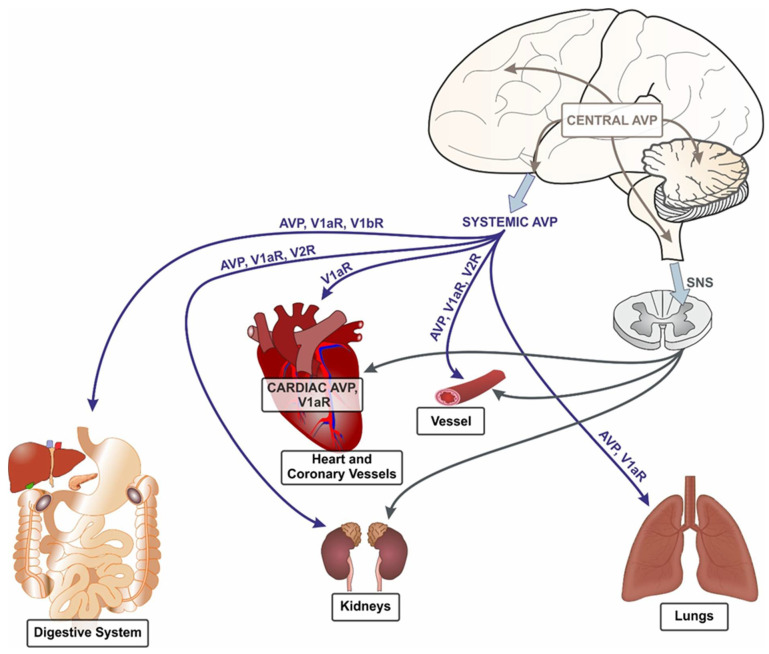
Components of the vasopressin system involved in the regulation of blood flow in the brain, heart, vessels, kidney, lungs and digestive system. AVP—arginine vasopressin; SNS—sympathetic nervous system; V1aR—vasopressin V1a receptors; V1bR—vasopressin V1b receptors; V2R—vasopressin V2 receptors.

**Figure 2 ijms-23-14414-f002:**
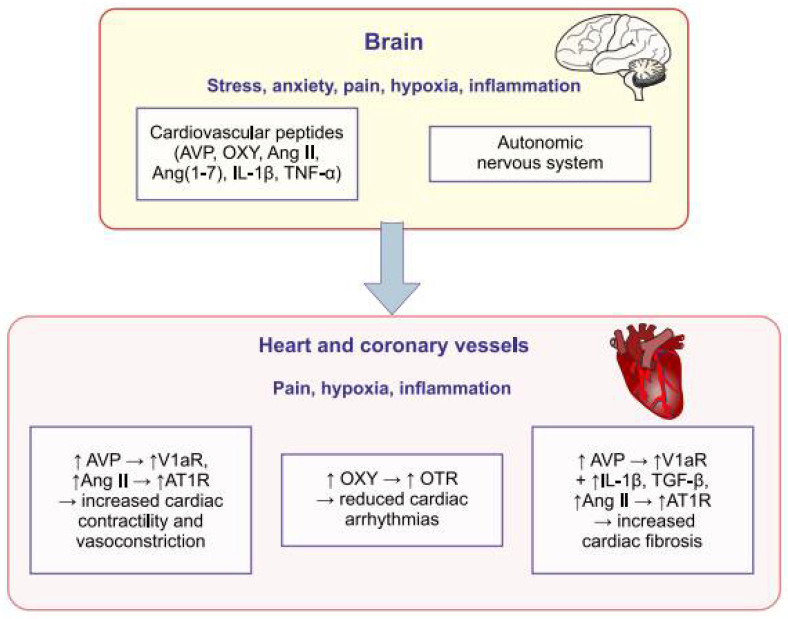
Interaction of vasopressin, angiotensins, oxytocin and cytokines with the autonomic nervous system in the regulation of cardiovascular parameters. Stress, anxiety, pain, hypoxia and inflammation enhance the release of vasopressin, oxytocin, angiotensins and cytokines in the brain, heart and vessels, and these peptides regulate the function of the heart and vessels through central and intracardiac effects. Ang II—angiotensin II; Ang (1–7)—angiotensin-(1–7); AT1R—AT1 receptors, AVP—arginine vasopressin; IL-1β—interleukin 1-β; OTR—oxytocin receptors; OXY—oxytocin; TGF-β—transforming growth factor β; TNF-α—tumor necrosis factor α; V1aR—vasopressin V1a receptors. → sequence of events, ↑- increased activation, ↓ - decreased activation.

## Data Availability

Not relevant.

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
