# Peer review of "The Heart as a Target of Vasopressin and Other Cardiovascular Peptides in Health and Cardiovascular Diseases"

_ijms, 2022, doi:10.3390/ijms232214414_

Round 1

Reviewer 1 Report

1)    Overall, the English grammar used throughout the article could be improved – hence I would recommend the use of services that specialise in correcting English grammar

2)    This review (esp. Intro) was filled with numerous very vague and ambiquous statements, that need to be more specific e.g. “AVP secretion is potently affected” – should indicate increase or decrease; “enhanced release of AVP significantly influences the cardiovascular regulation” – should once again define “influences” as up or down.

3)    Introduction: 2nd paragraph – the heart and vessels received excitatory information from ‘post’-ganglionic  nerves, not pre-sypnatic?

4)    Intro – what is meant by “walls of the cardiovascular system”?

5)    Figure 1 legend requires an expanded description and explanation. 

6)    Define ‘VPS’

7)    It is often difficult to differentiate whether the effect of AVP on it’s target tissue or system is a direct effect of AVP, or whether the effects are secondary or downstream of AVP’s action upstream.  E.g. “AVP has significant impact on magnitude of the cardiovascular and behavioral responses to stress”

8)    The section on Pain, Stress and Vasopressin seemed quite isolated and not necessarily part of this review.

9)    One should always avoid making claims that are not supported by statistical evidence e.g. “comparing action of epinephrine and vasopressin in human patients with ventricular fibrillation and cardiac arrest showed [some tendency] for better resuscitation in patients treated with AVP than in those treated with epinephrine but the groups of patients were relatively small and the [differences were not significant]”

10)Vasopressin and Angiotensins – it was unclear whether the authors were stating that Ang causes cardiovascular disease, or whether CVD alters the expression of Ang and it’s receptors, which further exacerbates CVD. E.g. the authors claim that the [experimental] aortic constriction was caused by overstimulation of AT1R. But the constriction was experimentally induced!

11) Vasopressin and Oxytocin – this section really should highlight that OT is released from key hypothalamic regions e.g. SON and PVN, which is very similar to AVP, which ties in with their comparable interactions.

12)Given the content of the abstract, it is surprising that the authors do not provide a more comprehensive discussion about AVP and the ANS. In particular, the close interaction between AVP, OT and ANS.

13) The sections describing AVP and cytokines and Insulin and glucagon are very brief; to the point that it is unclear what relevance they have in the review article as a whole.  

Author Response

Reviewer 1

Thank you very much for all valuable recommendations. The text has been corrected and altered parts of the manuscript have been shadowed in yellow.

Specific recommendations

  1. Overall, the English grammar used throughout the article could be improved – hence I would recommend the use of services that specialize in correcting English grammar.

Answer:

The English grammar and spelling were checked and I hope the errors are removed.

  1. This review (esp. Intro) was filled with numerous vary vague and ambiguous statements, that need to be more specific e.g. “AVP secretion is potently affected” – should indicate increase or decrease; “enhanced release of AVP significantly influences the cardiovascular regulation” – should once again define “influences” as up or down.

Answer:

The text has been analyzed and I hope the ambiguities are removed, but please, note that frequently the same compound may act I two directions, depending on the concentration or interaction with other compounds.

  1. Introduction: 2nd paragraph – the heart and vessels received excitatory information from ‘post’-ganglionic nerves, not presynaptic?

Answer:

The sentence is now altered, so that it indicates main components of the sympathetic innervation.

  1. Intro – what is meant by “walls of the cardiovascular system”?

Answer:

“walls of the cardiovascular system are replaced by “the heart and vessels”

  1. Figure 1 legend requires an expanded description and explanation.

Answer:

The legend of the Figure 1 is now expanded.

  1. Define VPS

Answer:

VPS is defined in the section 2 “General Overview of Vasopressin System”, first paragraph

  1. It is often difficult to differentiate whether the effect of AVP on it’s target tissue or system is a direct effect of AVP, or whether the effects are secondary or downstream of AVP’s action upstream. E.g. “AVP has significant impact on magnitude of the cardiovascular and behavioral responses to stress”

Answer:

  1. The whole text has been checked and I hope that all ambiguities have been removed.

The section on Pain, Stress and vasopressin seemed quite isolated and not necessarily part of this review.

Answer:

I did not decide to remove the section “role of vasopressin in pain and stress” because pain and stress are frequently present in cardiovascular diseases and both are important regulators of vasopressin release.

  1. One should always avoid making claims that are not supported by statistical evidence e.g. “comparing action of epinephrine and vasopressin in human patients with ventricular fibrillation and cardiac arrest showed [some tendency] for better resuscitation in patients treated with AVP than in those treated with epinephrine but the groups of patients were relatively small and the [differences were not significant].

Answer:

We agree with the comment of the Reviewer and the paragraph containing the sentence “Early  studies… were not significant” is now  removed.

  1. Vasopresin and Angiotensins – It was unclear whether the authors were stating that Ang causes cardiovascular diseases, or whether CVD alters the expression of Ang and it’s receptors, which further exacerbates CVD. E.g. that authors claim that the [experimental] aortic constriction was caused by overstimulation of AT1R. But the constriction was experimentally induced!

Answer:

The speculative part of the sentence “which suggested that the aortic constriction resulted from overstimulation of AT1R was removed.

  1. Vasopressin and Oxytocin – the section really should highlight that OT is released from key hypothalamic regions e.g. SON and PVN, which is very similar to AVP, which ties in with their comparable interactions.

Answer:

Information that the hypothalamic neurons are the main source of oxytocin has been introduced into the first paragraph of the section “Vasopressin and Oxytocin”

  1. Given the context of the abstract, it is surprising that the authors do not provide a more comprehensive discussion about AVP and the ANS. In particular, the close interaction between AVP, OT and ANS.

Answer:

The information about interactions of vasopressin and oxytocin with the autonomic nervous system is present in the third paragraph of the Introduction.

  1. The sections describing AVP and cytokines and Insulin and glucagon are very brief; to the point that it is unclear what relevance they have in the review article as a whole.

Answer:

The section concerning cytokines has been expanded in order to better explain their importance for the cardiovascular regulation and associations with vasopressin.  The section “Vasopressin, insulin and glucagon” has been removed and only some brief information is introduced to the section “Summary and conclusion”

Reviewer 2 Report

This current review article entitled "Heart as a target of vasopressin and other cardiovascular peptides in health and cardiovascular diseases", by Szczepanska-Sadowska is a good read. Author focused on the fuction of vasopressin (AVP) and some other cardiovascular peptides such as angiotensins, oxytocin, cytokines, insulin in the regulation of the cardiovascular system in health and cardiovascular disorders. This review also explains how vasopressin significantly interacts with the autonomic nervous system (ANS) and other cardiovascular peptides. This review also put a light on interaction of AVP with ANS and other CP during hemodynamic disturbances may have significant impact on responsiveness of the cardiovascular system to vasopressin in cardiovascular disorders. This review is well written and addresses a research topic of great interest. However, this reviewer has certain suggestions that would help produce a more comprehensive overview of the topic:   

Comments:

1. The English of manuscript can be polished (minor) and there are few typo errors in the manuscript that can be checked.

2, At least one illustrative figure may be provided as to highlight the summary of this study.

3, The authors should cross-check all abbreviations in the manuscript. Initially, define in full name followed by abbreviation.

4, In the section Vasopressin and Cytokines, author can add a paragraph on role of immune cells during myocardial infarction or heart failure. Some recent article can be used to put emphasis on immune cells role in cardiovascular diseases such as https://doi.org/10.3389/fcvm.2022.992653; PMID: 33396359; https://doi.org/10.1016/j.jacbts.2022.05.005; PMID: 34043424; PMID: 32998408; PMID: 35730443; PMID: 34119620; PMID: 33612829.

Author Response

Reviewer 2

Thank you very much for your valuable comments and suggestions. The text has been checked and corrected parts of the manuscript have been shadowed in yellow.

Specific recommendations

  1. The English of manuscript can be polished (minor) and there are few type errors in the manuscript that can be checked.

Answer:

The grammar and spelling errors have been removed.

  1. At least one illustrative figure may be provided as to highlight the summary of this study.

Answer:

According to the suggestion Figure 2 has been introduced to the section “Summary and Conclusion”.

  1. The authors should cross-check all abbreviations in the manuscript. Initially, define in full name followed by abbreviation.

Answer:

The abbreviations in text and description of figures have been cross-checked.

  1. In the section Vasopressin and cytokines, author can add a paragraph on role of immune cells during myocardial infarction or heart failure. Some recent article can be used to put emphasis on immune cells role in cardiovascular diseases such as “…”

Answer:

Thank you very much for your suggestions. The text in the section Vasopressin and Cytokines is now enriched by new information and new references have been introduced to this part of the manuscript and to the section References.